# Role and Functional Differences of HKT1-Type Transporters in Plants under Salt Stress

**DOI:** 10.3390/ijms20051059

**Published:** 2019-03-01

**Authors:** Akhtar Ali, Albino Maggio, Ray A. Bressan, Dae-Jin Yun

**Affiliations:** 1Department of Biomedical Science & Engineering, Konkuk University, Seoul 05029, Korea; gultkr@yahoo.com; 2Department of Agriculture, University of Naples Federico II, Via Universita 100, I-80055 Portici, Italy; almaggio@unina.it; 3Department of Horticulture and Landscape Architecture, Purdue University, West Lafayette, IN 47907-2010, USA; bressan@purdue.edu

**Keywords:** abiotic stresses, high salinity, HKT1, halophytes, glycophytes

## Abstract

Abiotic stresses generally cause a series of morphological, biochemical and molecular changes that unfavorably affect plant growth and productivity. Among these stresses, soil salinity is a major threat that can seriously impair crop yield. To cope with the effects of high salinity on plants, it is important to understand the mechanisms that plants use to deal with it, including those activated in response to disturbed Na^+^ and K^+^ homeostasis at cellular and molecular levels. HKT1-type transporters are key determinants of Na^+^ and K^+^ homeostasis under salt stress and they contribute to reduce Na^+^-specific toxicity in plants. In this review, we provide a brief overview of the function of HKT1-type transporters and their importance in different plant species under salt stress. Comparison between HKT1 homologs in different plant species will shed light on different approaches plants may use to cope with salinity.

## 1. Introduction

Plants are sessile organisms, which are continuously challenged by various biotic and abiotic environmental stresses, such as soil salinity, extreme temperatures, drought, nutrients deficiency or pathogen attack. These stresses have a tremendous impact on agricultural crops, reducing their potential yields by more than half [1]. Soil salinization is one of the most serious causes of stress for world’s agriculture, and it is progressing in most agricultural regions [2,3,4]. In saline soils, the ability of plants to grow and complete their life cycle can be severely compromised. Increased salinity leads to cytosolic osmotic stress and sodium ion specific toxicity that exert a combined inhibitory effect on physiological, biochemical and developmental pathways [5,6,7]. To deal with toxic levels of Na^+^, plants may restrict Na^+^ influx, compartmentalize Na^+^ to vacuoles and/or mobilize the un-avoidable influx of Na^+^ outside the cell and/or in different regions/organs of the plants [8,9,10,11]. In addition, the ability to take up K^+^, a plant essential nutrient, is also crucial under salinity stress [12,13,14]. 

In dealing with the potentially detrimental effects of Na^+^, sodium transporters play a pivotal role in plant protection in saline environments. These include antiporters that extrude Na^+^ from root cells and/or re-distribute Na^+^ throughout different tissues (Salt-Overly-Sensitive or SOS pathway) so as to reduce toxicity in critical cellular regions and reestablish to some degree water homeostasis [15,16,17]. Symporters, known as HKT1-type transporters (high-affinity potassium transporter1) also contribute to Na^+^ detoxification by retrieving/diverting Na^+^ from the xylem stream to protect the shoots from Na^+^ toxicity [18,19,20,21]. The function of this mechanism is to confine toxic ions to the roots, thus protecting above ground tissues from damage [8,9,22]. The critical role of HKT1 transporters under salt stress has been well characterized in a number of plant species including the model plant *Arabidopsis*, wheat, rice, sorghum, tomato, as well as in extremophile models such as *Eutrema parvula* and *Eutrema salsuginea* [8,23,24,25,26,27,28]. HKT1-type transporters mediate the balance between Na^+^ and K^+^ ions under salt stress, a function that has recently been reported also for HKT1 transporters in monocots [12,13,14,29]. In cereal crops such as wheat and rice, which contain multiple *HKT1*-type genes, some members of this transporters family have been identified as key components of plant salt stress tolerance [12,30,31].

## 2. Na^+^ Homeostasis

There are two main drawbacks of salt stress: osmotic stress and ion imbalance. Osmotic stress is caused by a decrease of the available water in the soil due to a reduced osmotic potential which makes more difficult for a plant to extract water [32]. Ion imbalance is mostly caused by excessive accumulation of Na^+^ ions, which can inhibit normal cellular functions [33]. To achieve protection against high salinity, plants need to activate mechanisms that regulate both Na^+^ uptake and homoeostasis [34,35]. The orchestrated distribution of Na^+^ ions throughout the entire plant body represents one crucial activity to keep Na^+^ away from sites of metabolism [36]. Moreover, the management of Na^+^ must be balanced against the control of specific ion toxicity and the uptake of K^+^, which is essential for normal plant growth and development [36]. High Na^+^/K^+^ ratios in the cytosol are toxic to plants, inhibiting various processes such as K^+^ absorption, vital enzyme reactions, protein synthesis and photosynthesis [5,37]. In order to control the adverse effects of salt stress, plants have evolved various adaptive mechanisms to control ion homeostasis regulated by several proteins, working alone or in a group. Among them, HKT1-type transporters regulate sodium homeostasis by keeping a balance between Na^+^ and K^+^ in the cytoplasm [14,19,21,22,38]. To further explain and sheds light on the role of HKT1-type transporters, we will briefly discuss their discovery and classification based on their cation selectivity.

## 3. Discovery of HKT1-Type Transporters

HKT1-type transporters have an important role in mediating the distribution of Na^+^ within the plant by a repeated pattern of Na^+^ removal from the xylem, particularly in the roots, so that the amount of Na^+^ reaching the shoot becomes more easily manageable [20,21,39]. Since the discovery of HKT1 in the early 90s [8,40], many more HKT transporter homologs from other species and with different cation transport properties have been isolated, which opened a new area of salt stress signaling in plants [7,23,25,27,28,41,42,43,44,45,46,47,48]. Interestingly, HKT1-type transporters from various species received the same name independently of their specific transport characteristics. For example, HKT1 from wheat was named TaHKT2;1 whereas its homolog from *Arabidopsis* was named AtHKT1. Nevertheless, the cation transport properties of these two HKT1 transporters are different from each other, AtHKT1 is a Na^+^ transporter whereas TaHKT2;1 is a K^+^/Na^+^ symporter [8,23]. Subsequently, due to their different cation selectivity, HKT1-transporters have been divided into different classes based on their cation transport properties [30,49].

## 4. Classification HKT1-Type Transporters

Homologs of *HKT1* genes and proteins have been identified in a number of plant species, including *Arabidopsis*. Typically, their ion selectivity has been characterized in yeast and/or *Xenopus oocytes* [23,41,43]. Based on protein structure and ion selectivity, HKT1-type transporters have been divided into two sub-classes with differences in the amino acids of the first pore domain (PD) of the protein as the main distinguishing feature. This was the basis of an international agreement for the nomenclature of HKT1-type transporters established back in 2006 [30]. Accordingly, members of class-1 contain a Ser (serine) residue at the first pore-loop domain (pA) and show higher selectivity for Na^+^ than K^+^ (Figure 1). In contrast, members of class-2 contain Gly (glycine) residues at the same position and are considered to function as Na^+^/K^+^ co-transporters (Figure 1) [24,30,50]. *Arabidopsis* contains a single copy *HKT1* gene, *AtHKT1,* that encodes for a member of class-1 and shows highly specific sodium influx when expressed in *Xenopus laevis oocytes* and *Saccharomyces cerevisiae* [23]. Monocots such as rice and wheat contain more than one copy of the *HKT1* gene and their coding proteins belong to both class-1 as well as class-2 [3,12,24,25].

## 5. Role of HKT1-Type Transporters in Glycophytes and Halophytes

HKT1-type transporters play a crucial role in Na^+^ homeostasis. Knock-out of *HKT1* leads to NaCl sensitivity in *Arabidopsis* [19,20]. Homologs of *HKT1* have been isolated from different glycophytic species including wheat [8,40], *Arabidopsis* [19,21,22,23], rice [12,38,48], eucalyptus [43], barley [51], tomato [26], sorghum [27], strawberry [52], pumpkin [53], poplar [54]. The *Arabidopsis* genome contains a single *HKT1* gene that encodes for AtHKT1, a member of class-1 transporters. AtHKT1 acts as a high-affinity selective Na^+^ transporter in heterologous systems such as *Xenopus oocytes* and yeast [19,23]. AtHKT1 has been shown to retrieve Na^+^ from the xylem stream to reduce its transport and accumulation to the shoots [19,20]. This process prevents Na^+^ toxicity in the shoots through recirculation of Na^+^ to the roots from which it could be exported again [8,9,22]. On the other hand, members of class-2 transporters contribute to maintain a balanced Na^+^/K^+^ ratio in the cytoplasm under salt stress. The mode of action of class-2 transporters depends on the external Na^+^ concentration. *TaHKT1* from wheat, a member of subclass-2, at low concentrations of Na^+^ works as Na^+^/K^+^ symporter, but at high concentration of Na^+^ TaHKT1 act as a Na^+^ uniporter [8]. It has been recently demonstrated, via transgenic analysis, that over-expression of *AtHKT1* contributes to maintain optimal K^+^/Na^+^ in tobacco plants and to improve plant biomass under salt stress [55]. However, different modifications of HKT1 transporters may also cause variations of leaf Na^+^ exclusion and salt tolerance in maize. Therefore, the exact mechanism through which HKT1 transporters confer salt tolerance deserves further attention. New insights in glycophytes have been obtained by comparative analysis of cultivated plants with wild relatives. In rice, different salt-tolerance during seed germination and seedling vegetative growth in weedy and cultivated plants has been associated to variants of *HKT1*-mediated transport and regulatory mechanisms that affect Na^+^/K^+^ balance [56].

Considering the activity of *HKT1* genes in glycophytic species, their functions in naturally salt-tolerant plants (halophytes) was also investigated [6,13,57]. Halophytes also control Na^+^ toxicity based on efflux and re-distribution of Na^+^ ions into various tissues to reduce its toxicity in specific plant organs and on Na^+^ sequestration in the vacuole [13,58]. The *Arabidopsis* close relative *Eutrema salsuginea* (previously *Thellungiella halophila*) is a model halophyte [6,7,13]. *E. salsuginea*’s genome has been recently sequenced and it provides a resource to characterize the function of different genes in this species [11,59]. Although the precise nature of mechanisms that regulate halophytism is not fully understood [6,13,60,61], much progress has been made based on a comparative analysis of halophyte with glycophytes. Similar to glycophytes, halophytes also rely on genes coding for salt overly sensitive (SOS), vacuolar Na^+^/H^+^ antiporter (NHX) and sodium transporter (HKT1) proteins to cope with high salinity [7,28,59,60,62]. However, growing evidence indicates that these genes may have temporal and spatial expression patterns under normal and stress conditions that differentiate halophytes vs. glycophytes [7,11,28,58]. 

## 6. Functional Differences in Halophytic and Glycophytic HKTs

One close relative of *Arabidopsis* is the halophyte *Eutrema salsuginea* (previously *Thellungiella halophile* or *Thulengiella salsuginea*) [6,11,63]. *E. salsuginea*’s genome sequence is known and its juxta-positioning with the *Arabidopsis* genome provides a genetic blueprint that highlights similarities as well as differences between these genomes [59]. The genome of *E. salsuginea* includes three copies of *HKT1* genes in a tandem array [59]. Of the three *HKT1* homologs only *EsHKT1;2* is dramatically induced at the transcript level following salt stress [7]. When expressed in yeast, EsHKT1;2 shows high affinity for potassium whereas EsHKT1;1 more likely behaves as AtHKT1, with high specificity for sodium uptake [7]. Another *Arabidopsis* halophytic relative is *Eutrema parvula* (now *Schrenkiella parvula*). The genome of *Eutrema parvula* contains two *HKT1* genes, *EpHKT1;1* and *EpHKT1;2* [62]. *EpHKT1;2* is induced very rapidly upon salt stress [28]. All of these halophytic *HKT1* genes (three from *E. salsuginea* and two from *E. parvula*) contain a Ser residue at the selectivity filter in the first pore-loop domain and have therefore been grouped as class-1 transporters [30]. Members of class-1 HKT1-transporters lack the ability to uptake K^+^. However, both EsHKT1;2 and EpHKT1;2 possess conserved Asp (aspartate) residue in the second pore-loop domain [7] and show K^+^ uptake ability which makes them functionally different from other members of this class such as AtHKT1 (Figure 2).

Excess of [Na^+^] in the cytosol impairs the optimal cytosolic Na^+^/K^+^ ratio, which is recognized by the plant as K^+^ deficiency indicating that potassium homeostasis is important for plants during salt stress [7,13,38,64]. Induction of *HKT1* under K^+^ shortage would be detrimental if HKT1 was Na^+^ specific [7,24,28,65]. Under salt stress, when the cytosolic sodium concentration reaches a toxic level, plants activate high-affinity potassium transporters to re-establish an optimal [Na^+^]/[K^+^] cellular balance [12,14,27]. One example of high-affinity potassium transporters is EsHKT1;2 (and possibly EpHKT1;2). Down-regulation of *EsHKT1;2* in *E. salsuginea* leads to hypersensitive phenotypes under K^+^-deficient conditions (Figure 3 and Figure 4). Based on these findings and on the activation of EsHKT1;2 and EpHKT1;2 in response to high salinity, these genes can be considered major contributors to the halophytic nature of *E. salsuginea* and *E. parvula* [28]. 

## 7. Importance of Conserved Amino Acids in the 2nd Pore-Loop of HKT1 in Glycophytes and Halophytes

As discussed earlier, certain residues in the HKT transporters have a crucial role in the functioning of the transporter. Alignment of most published HKTs with ScTRK1 showed that both EsHKT1;2 and EpHKT1;2 contain conserved Asp residues in their second pore-loop domains (Asp207 and Asp205, respectively), (Figure 2) [14]. Yeast ScTRK1, a known high-affinity potassium transporter [66], also carries an Asp in the second pore-loop position (Figure 2). However, in most HKT1 proteins an Asn (asparagine) is present at the above said position (Asn211 in AtHKT1), while SlHKT1;1 and SbHKT1-4 both carry Ser residues (Ser264 and Ser277, respectively) [12,14,26]. These considerations were confirmed by showing that Asp207 and Asp205, could impart selectivity to subclass-1 HKT1 transporters. Asp207 to Asn207 in EsHKT1;2 and Asp205 to Asn205 in EpHKT1;2 were able to abolish potassium uptake and generate canonical subclass-1 Na^+^-selective transporters [7,28]. In addition, changing the Asn residue in the 2nd pore-loop domain of AtHKT1 to Asp (N211D) resulted in a transporter that resembled EsHKT1;2 with high affinity for potassium transport. More importantly, transgenic *Arabidopsis* plants expressing AtHKT1^N211D^ tolerate salt stress more effectively than the wild type AtHKT1 and show exactly the same phenotype as *EsHKT1;2-* or *EpHKT1;2*-overexpressing Arabidopsis plants [14,28]. This means that HKTs from dicots can be differentiated from each other with respect to their monovalent cation selectivity by the presence of either Asp or Asn residues in the 2nd pore loop domain.

## 8. Substitution of Conserved Residues in the Pore-Region Affects the Cation Selectivity of HKT1-Transporters

The cation selectivity of HKT1 transporters is convertible by exchanging a single amino acid. Ser in the 1st pore loop domain appears not to be the only essential amino acid favoring K^+^ uptake (at least in *Arabidopsis* and *Eutrema* species), but it possibly functions as a supporting residue. Nevertheless, Ser or Gly at the first pore-loop differentiates class-1 and class-2 HKT-transporters based on their Na^+^ or Na^+^/K^+^ co-transport activity. However, this hypothesis failed to differentiate EsHKT1;2 which, although it contains a Ser residue at 1st pore-loop region and is a member of class-1 transporters, it unexpectedly functions as a K^+^ transporter. In addition, EsHKT1;2 and EpHKT1;2 both contain a conserved Asp residue in the 2nd pore-loop region which is the key residue for their cation selectivity [14,28]. In contrast, it has been shown as indispensable the presence of an Asp (D) replacing an Asn residue (N211) to convert the Na^+^ uniporter AtHKT1 into a Na^+^/K^+^ symporter [14]. Furthermore, the replacement of Asp with Asn in the EsHKT1;2 protein abolishes its potassium transport properties and it converts EsHKT1;2 into a Na^+^ uniporter. Substitution of the corresponding Asn in AtHKT1 to Asp (N211D) confirmed the importance of this residue by expression in yeast cells, *Xenopus oocytes* and *Arabidopsis* (Figure 5) [14]. More recently, EpHKT1;2 was also shown to carry an Asp (D205) in the 2nd pore-loop domain. When Asp205 was substituted by Asn, EpHKT1;2 lost its ability to tolerate sodium stress in the presence of potassium [28]. According to these reports, HKT-type transporters possess several key amino acids which define their transport properties. In this regard, the presence of only Ser or Gly residues might not be sufficient to assign HKT-transporters to a specific class. 

## 9. Contribution of HKT1 in Plant Na^+^ Homeostasis and Salinity Tolerance

A balance between Na^+^ and K^+^ ions under salt stress is crucial for plant survival [64], but it is not clear how such balance can be established under conditions of (hyper-) accumulation of Na^+^ (and to some degree Cl^−^) leading to osmotic stress and ionic imbalance [67]. The localization of AtHKT1 in the xylem parenchyma cells appears to provide an answer because its activity can reduce the flux of Na^+^ to the shoot tip in the (for most plants extremely rare) conditions of excess Na^+^ in the root-zone. It is believed that high-affinity potassium transporters will be active during salt stress. The presence and stress-induced activity of Na^+^/H^+^ antiporters have also been shown [11]. Yet other transporters are active in partitioning Na^+^ into vacuoles which can act as ultimate sinks for sodium ions [68]. 

For plants that are exposed to an excess of Na^+^, the function of HKT1 isoforms seems to have changed from a distribution role that curtails Na^+^ flux throughout the plant into a supporting role as K^+^ transporters. The ability of *Thellungiella* species to maintain a low cytosolic Na^+^/K^+^ ratio in the presence of high salinity stress has been shown [69]. Suppression of *HKT1* expression in *E. salsuginea* by RNAi leads to hyper-accumulation of sodium in the shoots, reduced sodium content in the roots and, consequently, a disturbed Na^+^/K^+^ ratio in the plant. Shoot sodium hyper-accumulation brings salt sensitivity suggesting that *EsHKT1;2* in *E. salsuginea* is one of the major components of its halophytic behavior (Figure 3 and Figure 4). While the RNAi targeted all *EsHKT1* copies, it is likely that the phenotype of the RNAi lines mirrored the function of *EsHKT1;2*, which is the most highly expressed copy among the three tandem duplicated *HKT1*paralogs in the *E. salsuginea* genome [59].

Our recent findings showed that AtHKT1^N-D^, like the native EsHKT1;2 transporter, contributed to salt tolerance. This was demonstrated based on the phenotype of AtHKT1^N-D^ both in yeast and transgenic Arabidopsis lines [14]. More importantly the current generated by AtHKT1^N211D^ in *xenopus oocytes* were more similar to that of EsHKT1;2 rather than AtHKT1, indicating that enhanced uptake of K^+^ can reduce Na^+^ toxicity [26]. Molecular and structural studies of both AtHKT1 and EsHKT1;2 and their mutated versions further explained that HKT1-proteins contain different charge distributions at the pore site [14].

## 10. Concluding Remarks

Salinity tolerance in plants is a very complex process whose components are only partially known. A comparative analysis of halophytic and glycophytic systems has helped to understand the nature and function of critical genes in salt stress adaptation [11,19,29,70]. It has been shown that ThSOS1 and EsHKT1;2, and more recently EpHKT1;2 are essential determinants of the halophytic behavior of *E. salsuginea* and *E. pavula* [7,11,28]. Unlike their *Arabidopsis* counterparts, strong induction and activity of EsHKT1;2 [7] and EpHKT1;2 [28] under salt stress might suggest that the co-evolution of these ion transporters play a critical role in shaping halophytic lifestyles of *E. salsuginea* and *E. parvula*. Functional studies in *Arabidopsis* homologs of these ion transporters and genetic duplication in halophytes may help us to understand how multiple ion tolerance has been acquired to support survival in an environment characterized by high levels of Na^+^ [11,59,62,71,72]. Therefore, to better understand salt tolerance in crop plants, additional studies must be directed towards defining the regulatory mechanisms operating differently in glycophytic and halophytic HKT1 so to reconcile expression and protein location with the extremely high salt stress tolerance of these species. Improving our understanding of key functional mechanisms that halophytes use to cope with high salinity could help us in designing applications and strategies to improve salt stress tolerance in crop plants in the future.

## Figures and Tables

**Figure 1 ijms-20-01059-f001:**
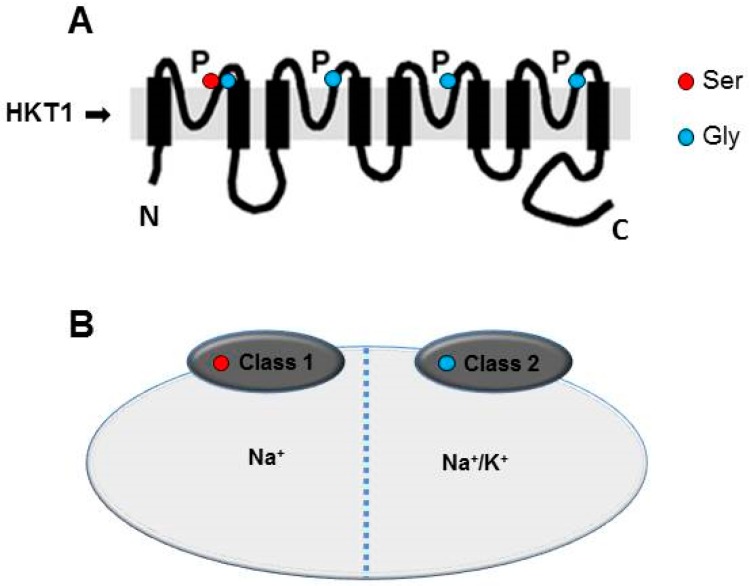
Classification and structure of HKT1 proteins. (**A**) Structural analysis of HKT1 protein containing Ser or Gly residues in their conserved regions. Members of class-1 transporters carry a Ser in the selectivity filter position while on the same position class-2 transporters contain a Gly residue. P denotes pore-loop domain while N and C indicate N-terminus and C-terminus of HKT-proteins. (**B**) Members of class-1 HKT1 that carry a Ser residue transport Na^+^ while members of class-2 that carry a Gly residue can transport both Na^+^ as well as K^+^

**Figure 2 ijms-20-01059-f002:**
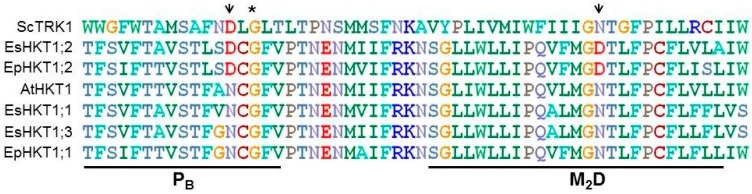
Sequence comparison of HKT homologs from *Arabidopsis, E. salsuginea* and *E. parvula*. Amino acid sequences in the second pore loop region (PB) and the adjacent transmembrane domain (M2B) are aligned by clustalw (http://www.ebi.ac.uk/Tools/msa/clustalw2/). The conserved Gly residues in the PB region [49] are indicated by asterisks. The Asp residues specific for EsHKT1;2 (D207) and EpHKT1;2 (D205) are indicated by arrows.

**Figure 3 ijms-20-01059-f003:**
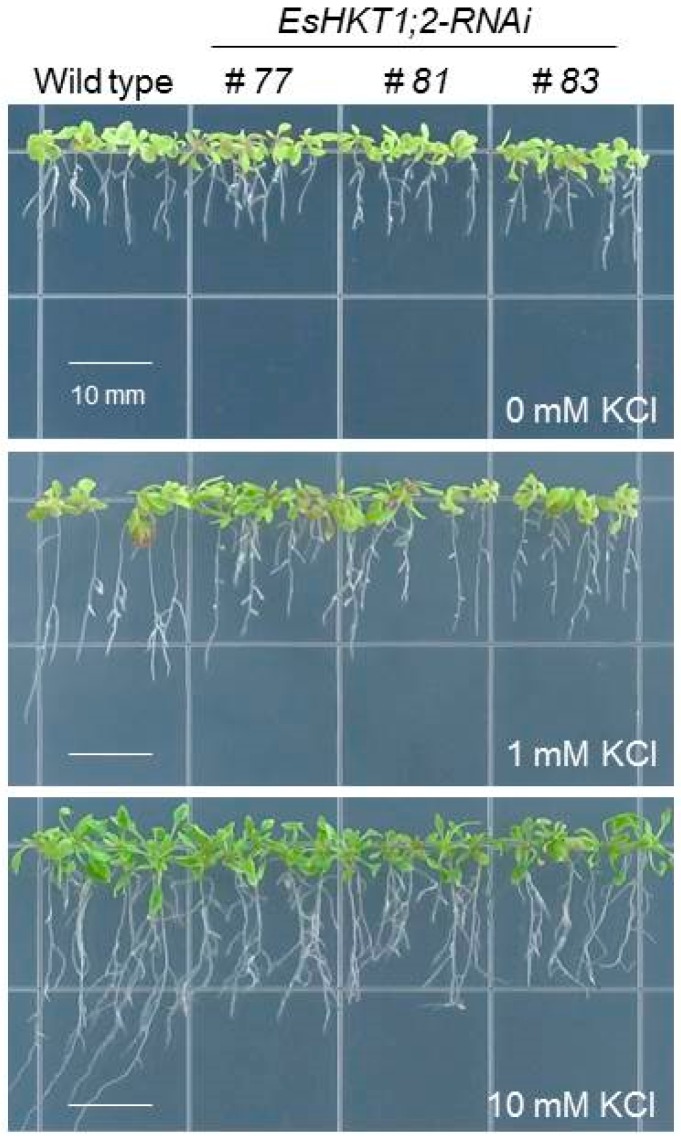
*EsHKT-RNAi* plants are sensitive to low K^+^-limiting conditions. Wild type and knock-down lines of *EsHKT1;2* (*EsHKT1;2-RNAi*) were grown on MS-medium for 10-days and then transferred to K^+^-deficient media with 0, 1 and 10 mM KCl (see Ali et al. 2012 for the detailed methodology) and allowed to grow for further 10-days. *EsHKT1;2-RNAi* lines were more sensitive to K^+^-limiting conditions as compared with the wild type Control. A gradual increase of K^+^ concentration greatly promotes the growth of wild type plants, whereas *EsHKT1;2-RNAi* lines were still sensitive. This result demonstrates the crucial role of EsHKT1;2 for K^+^-uptake.

**Figure 4 ijms-20-01059-f004:**
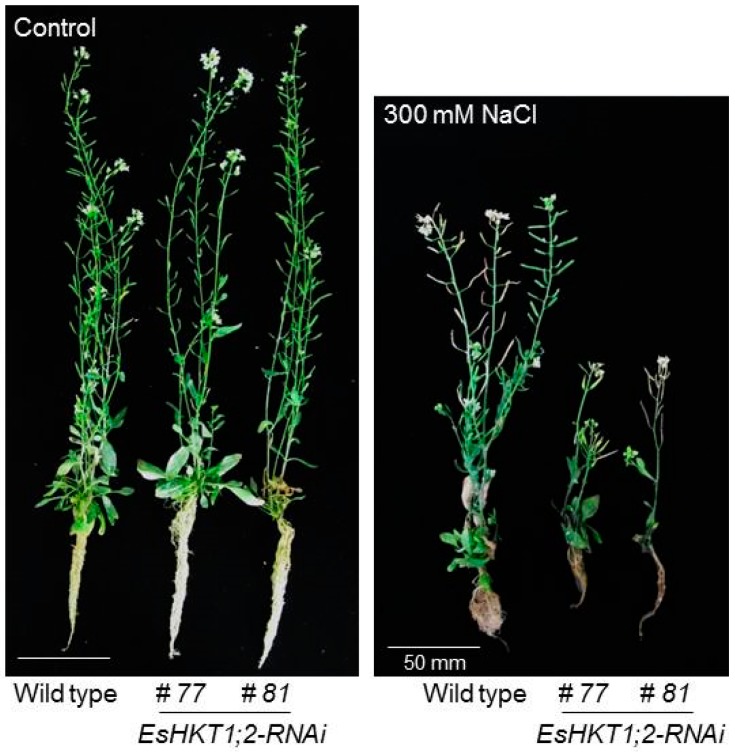
*EsHKT-RNAi* plants are sensitive to salt stress. Wild type and knock-down lines of *EsHKT1;2* (*EsHKT-RNAi*) were grown on MS-medium for 2-weeks and then transferred to inert soil (porous soil, see Ali et al. 2012 for details) and grown for further 3-weeks. Plants were then treated with 300 mM NaCl for another 3-weeks period, twice a week (control represents untreated plants). *EsHKT-RNAi* lines were more sensitive to salt stress as compared with wild type Control.

**Figure 5 ijms-20-01059-f005:**
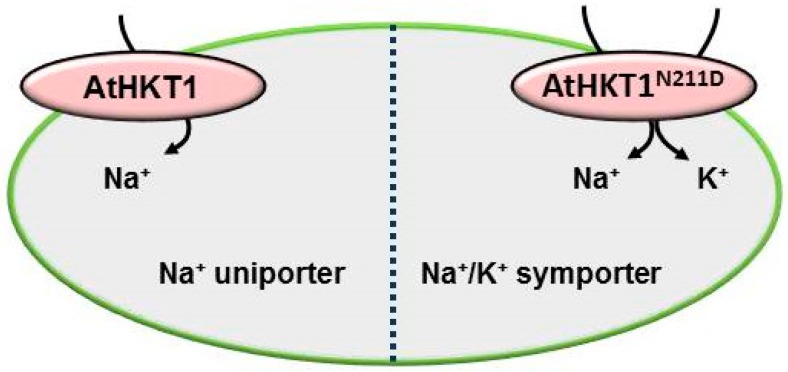
Functional properties of AtHKT1 and AtHKT1^N211D^ and differential selectivity for Na^+^ and K^+^ based on the Asn/Asp variance in the pore region. Wild type AtHKT1 is a sodium uniporter and does not confer salt stress tolerance. An altered version of AtHKT1 with a mutation of the Asn to Asp (AtHKT1^N211D^) is also able to uptake potassium and confers salt stress tolerance. It has already been shown by Ali et al. 2016 that *athkt1-1* plants complemented by *AtHKT1^N^**^211D^* showed higher tolerance to salt stress than lines complemented by the wild type *AtHKT1*. Thus, the introduction of Asp, replacing Asn, in HKT1-type transporters established altered cation selectivity and uptake dynamics.

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
