# Peer review of "Role and Functional Differences of HKT1-Type Transporters in Plants under Salt Stress"

_ijms, 2019, doi:10.3390/ijms20051059_

Reviewer 1 Report

HKT1 transporters are essential for plant response and adaptation to salinity therefore, the review on this topic emphasizing their role in plants sensitive (glycophytes) and in plants tolerant (halophytes) to salt stress are of interest to scientists who work on plant response to abiotic stresses, especially salt stress.

However, in my opinion the review is far from being perfect. My biggest concern here is very limited information on the role HTK1-type transporters.

-Titles of the chapters do not reflect their content. 

For example in the Chapter 3 “Discovery of HKT1-type transporters” there is only half of the sentence concerning the discovery of HKT1 transporters and this information is not fully true. Actually, to my knowledge Schachtman and Schroeder have discovered the first HKT1 transporter in 1994. Rublio et al. published their discovery one year latter in 1995. The authors of the manuscript did not provide even one sentence regarding these discoveries. Both papers have been published in very respective scientific journals (Nature and Science) and they should be getting more attention in this specific chapter.

In the chapter 5 “Role of Na+ transporters in halophytes”, in fact the role of Na+ transporters has not been described. The authors only mentioned  (lines 117-120) “Considering the strong correlation between the presence and activity of class-1 HKT1 genes in salt sensitive (glycophytic) species, it seemed important to investigate HKT1 functions in salt-tolerant (halophytic) plants. Conceivably, the extreme tolerance to often high levels of Na+ by halophytes suggests specialized mechanisms protecting them against high salinity “ 

Moreover, in lines 122-123, the authors wrote, “Halophytes, as far as we know, include genes coding for the SOS, NHX and HKT1 proteins that regulate halophytism” 

In my opinion this not a special feature of halophytes, since SOS1, NHX, HKT1 transporters are encoded not only in halophytes but also glycophytes genomes. However, important are differences in their expression levels and the transporters activity in halophytes and glycophytes. Unfortunately, the authors do not describe these issues. In this regard in this chapter is only one sentence (lines 124-126)„However, indications exist concerning differences in how these genes might be expressed under normal and stress conditions in halophytes compared to glycophytes (Gong et al. 2005; Oh et al. 2009; Ali et al. 2012, 2018). 

-In my opinion the chapter on the role of Na+ transporters in glycophytes is missing. It should be included in the manuscript.

- Chapter 4. “Classification HKT1-type transporters”

The chapter will improve by adding some figures e.g., (-) figure presenting the amino acid structure of the first pore domain (PD) of different plant species including halophytes and glycophytes, with marked amino acids specific for class-1 and class-2;

(--) phylogenetic tree of plant HKT1 type transporters.

--Figure 4 is oversimplified. Looking at the figure, the reader can make an assumption that AtHKT1 is a negative regulator of the plant tolerance to salinity, whereas it is not true.

Author Response

Response to Reviewer 1 – RC: reviewer comment; OR: Our reply

RC: HKT1 transporters are essential for plant response and adaptation to salinity therefore, the review on this topic emphasizing their role in plants sensitive (glycophytes) and in plants tolerant (halophytes) to salt stress are of interest to scientists who work on plant response to abiotic stresses, especially salt stress. However, in my opinion the review is far from being perfect. My biggest concern here is very limited information on the role of HKT1 transporters.

OR: We provide now in section 5 more info on the role of HKT1 transporters with addition of 7 more references. However, we would like to emphasize that the overall scope of this review was not to list all the papers on HKT1 transporters and relative findings, as many published excellent reviews have already done, but only to refer to those and bring the attention of the reader on functional differences in the protein structures that may differentiate these transporters. In contrast to standard reviews on this topic we integrated it with our own unpublished data on this aspect (Figure 3 and Figure 4). But we agree with Reviewer 1 that the title of the manuscript could have been misleading and we changed it accordingly.

RC: Titles of the chapters do not reflect their content. For example in the Chapter 3 “Discovery of HKT1-type transporters” there is only half of the sentence concerning the discovery of HKT1 transporters and this information is not fully true. Actually, to my knowledge Schachtman and Schroeder have discovered the first HKT1 transporter in 1994. Rublio et al. published their discovery one year later in 1995. The authors of the manuscript did not provide even one sentence regarding these discoveries. Both papers have been published in very respective scientific journals (Nature and Science) and they should be getting more attention in this specific chapter.

OR: Thanks for pointing this out. We refer to both papers now (Schachtman and Schroeder and Rubio et al.). As mentioned in our previous answer, the purpose of this review was to add some new info to most published literature on this topic. However, following the request of reviewer 1, we did add a few more details in section 5.

RC: In the chapter 5 “Role of Na+ transporters in halophytes”, in fact the role of Na+ transporters has not been described. The authors only mentioned (lines 117-120) “Considering the strong correlation between the presence and activity of class-1 HKT1 genes in salt sensitive (glycophytic) species, it seemed important to investigate HKT1 functions in salt-tolerant (halophytic) plants. Conceivably, the extreme tolerance to often high levels of Na+ by halophytes suggests specialized mechanisms protecting them against high salinity.

OR: We added an entirely new section 5 addressing both glycophytes and halophytes.

RC: Moreover, in lines 122-123, the authors wrote, “Halophytes, as far as we know, include genes coding for the SOS, NHX and HKT1 proteins that regulate halophytism”. In my opinion this not a special feature of halophytes, since SOS1, NHX, HKT1 transporters are encoded not only in halophytes but also glycophytes genomes. However, important are differences in their expression levels and the transporters activity in halophytes and glycophytes. Unfortunately, the authors do not describe these issues. In this regard in this chapter is only one sentence (lines 124-126)„ However, indications exist concerning differences in how these genes might be expressed under normal and stress conditions in halophytes compared to glycophytes (Gong et al. 2005; Oh et al. 2009; Ali et al. 2012, 2018).

OR: The sentence and entire paragraph has been rearranged as suggested.

RC: In my opinion the chapter on the role of Na+ transporters in glycophytes is missing. It should be included in the manuscript.

OR: Section 5 includes now a list of references on transporters in glycophytes with 7 new references updated to 2019.

RC: Chapter 4. “Classification HKT1-type transporters” - The chapter will improve by adding some figures e.g., (-) figure presenting the amino acid structure of the first pore domain (PD) of different plant species including halophytes and glycophytes, with marked amino acids specific for class-1 and class-2; (--) phylogenetic tree of plant HKT1 type transporters.

OR: As requested, we included a figure (Figure 1) presenting the amino acid structure of the first pore-loop domain (PD).

RC: Figure 4 is oversimplified. Looking at the figure, the reader can make an assumption that AtHKT1 is a negative regulator of the plant tolerance to salinity, whereas it is not true.

OR: We agree that this may have been misleading. Figure 4 (now Figure 5) has been modified by replacing “salt sensitive” with “Na+-uniporter for AtHKT1” and “salt tolerant” with “Na+/K+ - symporter for AtHKT1N211D”.

Reviewer 2 Report

Please see the corrections in the attached PDF file. The corrections that needs be addressed are underlined or highlighted.  

Author Response

Response to Reviewer 2 – RC: reviewer comment; OR: Our reply

OR: We made all the suggested changes (indicated directly on the ms by Reviewer 2). Specifically, we corrected/accepted all the typos, editorial suggestions, word change and:

1)      We added one figure (Fig. 1) on the HKT protein structure;

2)      Expand abbreviated version of some genes when appropriate and as indicated by the reviewer;

3)      Include a reference at line 177 before Figure 2 (Ali et al., 2012);

4)      Include the length bar in figure 3;

5)      Maintain the same format when we refer to amino acids.

Reviewer 3 Report

This is a detailed review focus on function of HKT1-type transporters in halophytes and glycophytes. The importance of conserved AA regions was discussed, as well as the role of HKT1 in salinity tolerance. One comment is regarding the title. I think authors did not discuss about "soil salinity" much in glycophytes and halophytes in this review. So a concise title maybe more appropriate.   

Author Response

Response to Reviewer 3 – RC: reviewer comment; OR: Our reply

RC: This is a detailed review focus on function of HKT1-type transporters in halophytes and glycophytes. The importance of conserved AA regions was discussed, as well as the role of HKT1 in salinity tolerance. One comment is regarding the title. I think authors did not discuss about "soil salinity" much in glycophytes and halophytes in this review. So a concise title maybe more appropriate.  

OR: We changed the title to better reflect the content of the review.

Round  2

Reviewer 1 Report

The manuscript has been significantly improved. In my opinion only editing should be slightly corrected.